# Electronic control of gene expression and cell behaviour in *Escherichia coli* through redox signalling

Tanya Tschirhart[1], Eunkyoung Kim[1], Ryan McKay[1,2], Hana Ueda[1,3], Hsuan-Chen Wu[1], Alex Eli Pottash[1,2], Amin Zargar[2], Alejandro Negrete[4], Joseph Shiloach[4], Gregory F. Payne[1,2] & William E. Bentley[1,2]

The ability to interconvert information between electronic and ionic modalities has transformed our ability to record and actuate biological function. Synthetic biology offers the potential to expand communication 'bandwidth' by using biomolecules and providing electrochemical access to redox-based cell signals and behaviours. While engineered cells have transmitted molecular information to electronic devices, the potential for bidirectional communication stands largely untapped. Here we present a simple electrogenetic device that uses redox biomolecules to carry electronic information to engineered bacterial cells in order to control transcription from a simple synthetic gene circuit. Electronic actuation of the native transcriptional regulator SoxR and transcription from the P*soxS* promoter allows cell response that is quick, reversible and dependent on the amplitude and frequency of the imposed electronic signals. Further, induction of bacterial motility and population based cell-to-cell communication demonstrates the versatility of our approach and potential to drive intricate biological behaviours.

[1] Institute for Bioscience and Biotechnology Research, University of Maryland, 4291 Fieldhouse Drive, 5112 Plant Sciences Building, College Park, Maryland 20742, USA. [2] Fischell Department of Bioengineering, University of Maryland, 8228 Paint Branch Drive, 2330 Jeong H. Kim Engineering Building, College Park, Maryland 20742, USA. [3] Mathematics Department, University of Maryland, 4176 Campus Drive—William E. Kirwan Hall, College Park, Maryland 20742, USA. [4] Biotechnology Core Laboratory, National Institute of Diabetes and Digestive and Kidney Diseases, National Institutes of Health, 14 Service Road West, Bethesda, Maryland 20892, USA. Correspondence and requests for materials should be addressed to W.E.B. (email: bentley@umd.edu).

The exchange of information between electrons and ions has been a mainstay in a variety of biochemical applications for decades. Small molecules, however, represent a much wider 'repertoire' for biological information transfer, or 'molecular communication'. Gaining the ability to measure, disrupt or enhance these biomolecular signals would allow for development of advanced technologies to study and manipulate the biological environment. Specifically, molecular connectivity with electronics can benefit from the fact that electrochemical detection is sensitive, selective, cost-efficient and label-free in small volumes[1–3]. Such connectivity presents a unique opportunity to apply our knowledge of and control over electronic-device form and function to study biological systems[4], improve biosensors[2,5] and create wearable and implantable bio-hybrid devices[6–8].

Redox biomolecules have significant roles in a wide array of cellular functions, and present a means for electronically interceding with both native cell pathways and redox-sensitive engineered constructs[9–11]. Bioelectrochemical technologies such as microbial fuel cells (MFCs) and bioelectro-synthesis systems (BESs) use electrochemical techniques to interact with cellular redox processes and electron transport mechanisms to change or measure cellular behaviours. A plethora of literature exists on MFCs, where microbial communities metabolize organic compounds, resulting in production of electricity[12–14]. Conversely, BESs aim to electrochemically intercede with microbial metabolism for the production of various compounds of interest[15,16]. Electronic interrogation of biological systems with redox molecules has allowed for detection of changes in cell metabolic activity[17–19], redox state[20–22], toxicity[23] and other parameters[4]. Cells have been engineered for enhanced electron flow[24,25] and to allow for electronic detection of engineered cell activity[26,27]. Electronic signals translated through redox molecules also show controlled glucose consumption[28] and regulation of enzymatic activity[29]. The use of the above-mentioned and other bioelectrochemical methods will no doubt continue to have impactful applications in fields such as bioenergy, biotechnology, biosensing and biocomputing[30].

However, while the accomplishments above are impressive, they are limited in their cellular effects to those that are naturally responsive to changes in electron transfer or redox status. Linking electronic signals, through redox molecules, to engineered gene expression, opens the doors for electronically studying and controlling a much wider array of behaviours and thus the possibility of many additional applications. Such an electrogenetic device was previously explored in mammalian cells[31]. We advance this idea by working with *Escherichia coli*, a widely used synthetic biology chassis, and show circuit versatility and quick response times.

We use pyocyanin (Pyo) for gene induction and ferricyanide (Fcn) for response-amplification and electronic control to guide production of proteins that act as reporters or that otherwise direct cell function. Pyocyanin is secreted by *Pseudomonas aeruginosa* and is implicated in community organization, pathogenicity and interspecies behaviour[32–34]. To use pyocyanin as an inducer, we employed one of the best-characterized redox-responsive regulons in *E. coli*, the SoxRS regulon[32,35–37], which functions to sense and respond to oxidative stress. In *E. coli*, the SoxR protein contains an iron-sulfur cluster (2Fe-2S) that is maintained in a reduced form by NADPH-dependent enzymes[38]. When oxidized (for example, by redox-cycling drugs[32,38,39]), SoxR activates transcription of the SoxS protein from the P*soxS* promoter. The SoxS protein, in turn, regulates dozens of other genes, mainly with the aim of detoxifying the cell[40].

Studies of the mechanisms of redox-drug activation of SoxR show that conditions that promote cellular respiration increase expression from the P*soxS* promoter[32]. They suggest that this is due to increased electron flow through the respiratory machinery, which could allow increased re-oxidation of the redox drugs and SoxR activation. We worked from this hypothesis, and propose that using a redox molecule that acts as an electron acceptor and whose form we could electronically regulate would allow us to amplify the intracellular Pyo redox cycling that leads to SoxR-mediated transcription. We chose ferricyanide as our alternative electron acceptor. Ferricyanide (oxidized, Fcn(O)) and ferrocyanide (reduced, Fcn(R)) (with a standard potential, $E^0$, of $\sim +0.2$ V versus Ag/AgCl— silver/silver chloride) have been used for decades in studies of electron transport processes, where Fcn(O) reduction rates correlate with microbial respiratory activities[18,41,42].

Our method demonstrates electronic control of a native redox process to actuate gene expression. This bacterial electro-genetic device is simple, specific and versatile. We take advantage of the well-characterized native redox-response of the SoxRS regulon and proposed electron transport mechanisms so that minimal genetic 'rewiring' is required. Induction levels are controlled by varying either the applied electronic potential or its duration, and correlate to the measured charge through Fcn(O/R) redox form interconversion. We show that gene expression is functionally reversible on relatively short time scales (30–45 min) and that this allows for response 'ON'/'OFF' cycling. Additionally, we expand on this genetic circuit by demonstrating electronic induction of cell motility and by connecting electronically actuated cells to non-actuated cells via generation of the native signalling molecules associated with bacterial quorum sensing. Thus, electrons are converted to biological signalling molecules that, in turn, influence phenotype in otherwise unaffected cell populations. Importantly, the 'controlled' behaviours that our electrogenetic device controls are typically not responsive to such redox changes.

## Results

**Redox mediator effects on cells and gene expression.** Figure 1a provides a schematic representation of our approach. To test the effect of pyocyanin, Fcn(O), and Fcn(R), on gene expression and their interactions with cells we first carried out studies using chemical systems (for example, without electrodes; chemical structures are presented in Supplementary Fig. 1). These results, which for brevity are presented in the Supplementary Information, set the stage for electrode-based studies. We constructed plasmid pTT01, from the pBR322 vector, that includes the *soxR* gene and the overlapping divergent P*soxR* and P*soxS* promoters. The gene coding for the fluorescent reporter protein phiLOV[43], which can fluoresce in anaerobic conditions, was placed downstream of P*soxS* (Fig. 1b). We incorporated an ssRA[44] degradation tag—AANDENYADAS (DAS) on the C terminus of phiLOV in plasmid pTT01 (forming plasmid pTT03), which significantly increases protein degradation and thus results in an overall lower steady-state protein level, but also a more rapid return to baseline levels upon cessation of induction (denoted 'OFF'). All constructs, unless otherwise stated, were tested in the strain DJ901 ($\Delta soxRS$)[36]. This strain allowed for higher reporter levels, but cells with intact *soxRS* were still responsive (see 'Electronic actuation of bacterial motility' section and Supplementary Fig. 2). See Methods, Supplementary Fig. 3, and Supplementary Tables 1–3 for all plasmid and cell engineering information and sequences.

The addition of pyocyanin alone (0–10 µM) resulted in modest phiLOV expression (fluorescence increase from 200 to 500 au).

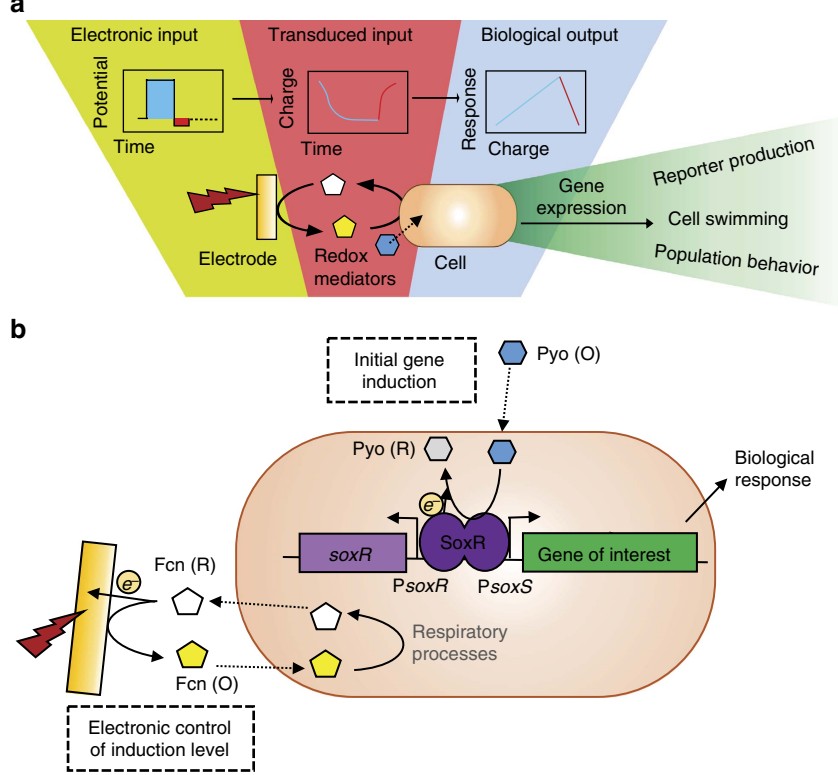

**Figure 1 | Electrogenetic device scheme.** (**a**) Device-mediated electronic input consists of applied potential (blue or red step functions) for controlling the oxidation state of redox-mediators (transduced input). Redox mediators intersect with cells to actuate transcription and, depending on actuated gene-of-interest, control biological output. (**b**) The electrogenetic device consists of the region encompassing the gene coding for the SoxR protein and the divergent overlapping P*soxR*/P*soxS* promoters. A gene of interest is placed downstream of the P*soxS* promoter. Pyo (O) initiates gene induction and Fcn(R/O), through interactions with respiratory machinery, allows electronic control of induction level. Fcn (R/O), ferro/ferricyanide; Pyo, pyocyanin. The oxidation state of both redox mediators is colorimetrically indicated (Fcn (O) is yellow pentagon; Fcn (R) is white pentagon; Pyo (O) is blue hexagon; Pyo (R) is grey hexagon). Encircled 'e⁻' and arrows indicate electron movement.

The addition of 5 mM of Fcn(O) amplified this pyocyanin-induced fluorescence ∼17-fold. Control cultures showed no increase in fluorescence (for example, Fcn (R) + Pyo or Fcn(O) only). See Supplementary Note 1 and Supplementary Fig. 4.

Additionally, phiLOV fluorescence increased with ferricyanide (0–25 mM) while pyocyanin was kept at 5 μM, in an apparent dose-dependent response (Supplementary Fig. 4). The results indicated the importance of the redox status of Fcn(O/R) since Fcn(O) but not Fcn(R) amplified pyocyanin-induced gene expression. Since this is the first study of this electrogenetic device, we performed the above and all following experiments anaerobically to exclude oxygen's interference with pyocyanin redox state and for better control of redox conditions. However, the system could be adapted for conditions that span a variety of oxygen gradients through further optimization. We discuss this and show preliminary data in Supplementary Note 2 and Supplementary Fig. 5.

Based on the above, we worked with 5 μM pyocyanin and 5 mM Fcn (O/R) for the remaining studies. At these levels, neither mediator significantly altered cell viability, though the combination did alter acetate production per glucose consumed (Supplementary Fig. 6 and Supplementary Note 3). We found that Fcn(O) reduction by cells depended on both the amount of cells and starting Fcn(O) concentration (Supplementary Fig. 7), consistent with above-mentioned literature regarding Fcn(O) use for respiratory activity measurement. As mentioned previously, others have proposed that redox-cycling drugs which oxidize SoxR and drive expression from the P*soxS* promoter

interact with the electron transport machinery[32]. We propose that in our system, after oxidation of SoxR, the now-reduced drugs are re-oxidized intracellularly when an electron acceptor is present. We provide corroborating evidence in Supplementary Fig. 8 and Supplementary Note 4, though we cannot rule out alternative mechanisms.

In sum, chemical studies demonstrated that Pyo induces *phiLOV* expression from the P*soxS* promoter and Fcn (O) but not Fcn (R) amplifies this expression in a dose-dependent manner.

**Electronic control of gene expression and dose–response.** The above results suggested the possibility for genetic induction *in situ* by applying electronic signals that provide negative charge (oxidation) to ensure both that PYO is oxidized and to increase Fcn(O) from Fcn(R), and positive charges (reduction) for subsequently halting gene induction through Fcn(O) reduction to Fcn(R) (Fig. 2a). We interconverted bulk Fcn (O/R) redox state electrochemically in a three-electrode set-up (Supplementary Fig. 9, Methods). In our system the $E^0$ of the Fcn (O/R) couple was about + 0.2 V (grey cyclic voltammogram in Fig. 2b, Supplementary Fig. 10a). For complete and quick bulk oxidation and reduction (<20 min, Supplementary Fig. 10b and c), we biased electrodes significantly more positively than the oxidation peak (+ 0.5 V for ∼ + 0.25 V peak) or more negatively than the reduction peak (− 0.3 V for ∼ + 0.1 V peak). We could use potentials closer to the peak potentials, but conversion

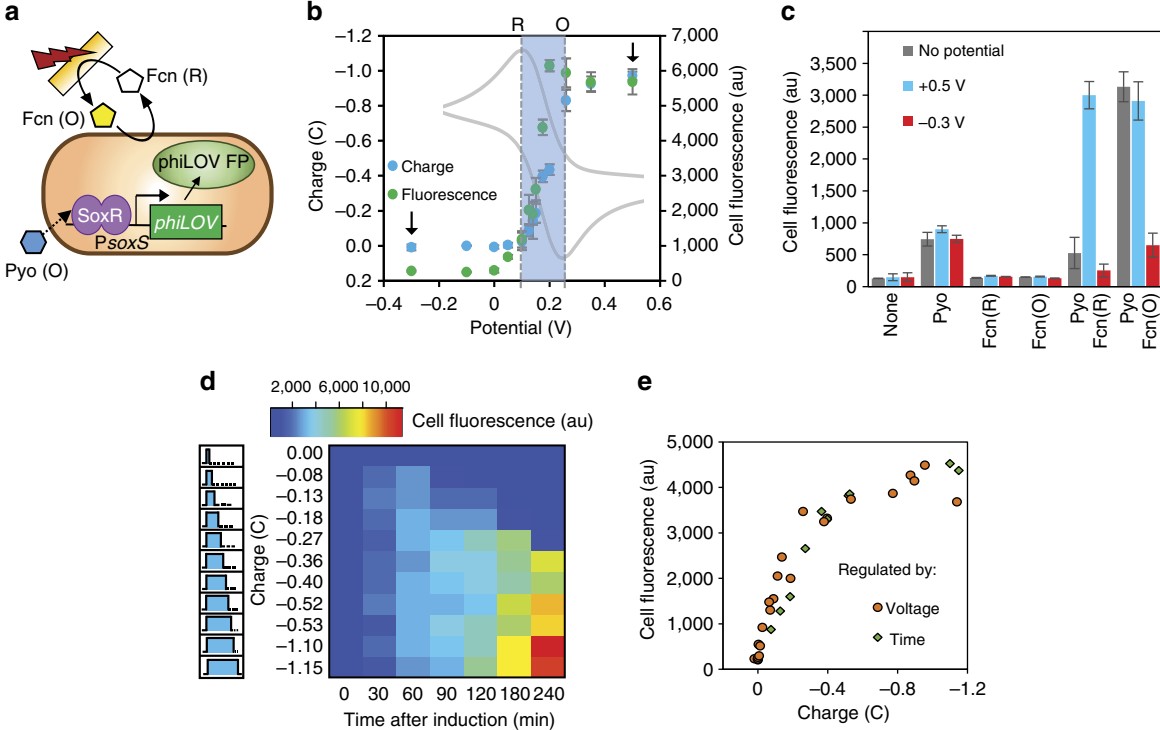

**Figure 2 | Electronic control of cell fluorescence. (a)** Schematic of electrogenetic device induction of the phiLOV fluorescent protein. *soxR* and P*soxR* omitted from schematic, but present. (**b**) Charge and average cell fluorescence resulting from applying the indicated potentials with Fcn (R) and Pyo. Grey cyclic voltammogram shows reduction (R) and oxidation (O) peaks of Fcn (R/O). Arrows indicate oxidizing ( + 0.5 V) and reducing ( − 0.3 V) potentials. (**c**) Cell fluorescence resulting from applied potential in the presence of indicated mediators. (**d**) Heat map showing cell fluorescence over time of samples induced with the indicated charges. Left panel indicates graphic representation of charge (area of shaded blue) increasing with application length of time of oxidizing potential ( + 0.5 V). (**e**) Overlay of cell fluorescence (averages over 4 h) from **b**,**d** and Supplementary Fig. 10 plotted against the applied charge. Error bars in **c** indicate s.d. of biological triplicates. Fcn (R/O), ferro/ferricyanide; Pyo, pyocyanin; V, Volts; C, Coulombs.

efficiency would suffer; conversely, higher voltages can generate unwanted reactive species. The measured charge (integrated current over time) correlated well with Fcn(O) absorbance (Fcn (O) is yellow) (Supplementary Fig. 10d) and repeated oxidation and reduction of the same solution did not degrade Fcn (O/R) (Supplementary Fig. 10e).

Correspondingly, in Fig. 2b, we show that varied electrode potential modulates Fcn(R) oxidation (charge) and cell (phiLOV) fluorescence. In these experiments, we applied different voltages to Pyo + Fcn(R) solutions with cells for 15 min, followed by incubation to allow for phiLOV accumulation, and flow cytometry measurements. Figure 2b shows the resulting total charge and average cell fluorescence levels at specific potentials. Three response ranges were observed based on the potential used. When we applied potentials more negative than the reduction peak, which did not promote significant Fcn(R) -> Fcn(O) conversion, charge and fluorescence outputs were negligible. Applied potentials between the reduction and oxidation peaks ( ∼ + 0.1 and + 0.25 V) resulted in proportionally more negative charge (partial Fcn(R) to Fcn(O) conversion) and increasing fluorescence. Potentials more positive than that of the oxidation peak resulted in a leveling off of charge and maximally induced cells. Based on these results, we confirmed + 0.5 V as our oxidizing potential (for Fcn (R) to Fcn (O) conversion) and − 0.3 V as our reducing potential (for Fcn (O) to Fcn (R) conversion) for future experiments. Control experiments confirmed that cells with both Pyo and Fcn(O) (either added or oxidized from Fcn(R)) were required for fluorescence amplification above pyocyanin-only levels, similarly to chemically induced experiments (Fig. 2c).

We tested whether Fcn(R) oxidized *in situ* by a constant oxidizing potential could amplify gene expression and whether increasingly negative charge, mediated by increasing duration (10–900 s), could elicit a dose-dependent response. A heat map depicts the cell fluorescence due to varied durations of applied + 0.5 V (Fig. 2d). Low charge (closer to zero) resulted in low cell fluorescence. For charges between zero and ∼ − 0.27 C, initial increases in fluorescence were followed by decreases. In these situations, the Fcn(R) amount converted to Fcn(O) was not sufficient to enable continued expression over the timeframe tested and ssRA-mediated phiLOV degradation brought the reporter quantity down. More negative charges than − 0.27 C resulted in higher Fcn(O) levels and continued increase in fluorescence for the length of the experiment (despite the ssRA-mediated degradation). A heat map with the corresponding cell fluorescence of induction with varied potentials is shown in Supplementary Fig. 11, and results are comparable.

In Fig. 2e, we show that the average fluorescence (via moving-window time average) increased with applied charge whether + 0.5 V was applied for varied lengths of time or potentials were varied but applied for 15 min. We found no significant differences between the two methods—highlighting that it was the applied electronic charge, not the voltage or its duration, which correlated with fluorescence. The response appeared linear until ∼ − 0.5 C. These experiments demonstrate a direct relationship between applied potential (electronic signal), resulting charge (Fcn(R) to Fcn(O) interconversion at the electrode), and average cell fluorescence, confirming electronic control of gene expression and defining the redox-based communication pathway.

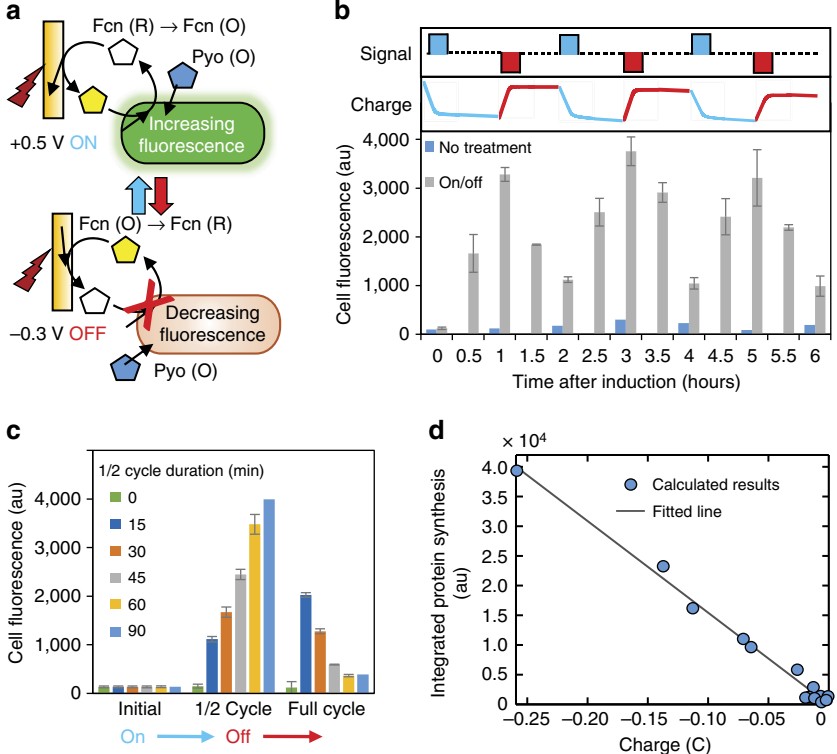

**Figure 3 | Electronic control of On/Off of fluorescence.** (**a**) Schematic of dynamic experiments with electronic signals to increase ('ON') or decrease ('OFF') fluorescence. (**b**) Fluorescence of cells from an extended culture cycled 'ON' and 'OFF' by potential applied in upper panel (signal; blue—'ON' and red—'OFF'). Charge is indicated in middle panel, with y axis ranges: 'ON' is 0 to − 2 C; 'OFF' is − 2 to 0 C. (**c**) Cell fluorescence after 'ON'/'OFF' cycles with the indicated durations. 'ON' potential applied at start and 'OFF' after ½ cycle measurement is taken. (**d**) Linearity between protein synthesis and charge. The value 'Integrated protein synthesis' represents the calculated accumulation of phiLOV fluorescence in the absence of degradation using the Matlab model. Error bars indicate s.d. of biological triplicates.

**Dynamic control of gene expression**. We wanted to take advantage of the dynamic electrochemical control of the redox state of Fcn(O/R) to drive overall reporter response 'ON' or 'OFF', characterized by increased protein production ('ON') or decreased protein production and quantity via the ssRA tag ('OFF'). We thus first tested the effects chemically by centrifuging and re-suspending cells in fresh media with different mediators to evaluate the genetic response from an 'ON' (Pyo + Fcn(O)) to 'OFF' (Pyo + Fcn (R)) transition (Supplementary Fig. 12a). In this situation *phiLOV* induction is reduced, the remaining protein (with ssRA tag, see Supplementary Fig. 12b) degrades, and the total fluorescence decreases. Repeated cycling of 'ON' to 'OFF' induction conditions at 1 h intervals showed corresponding fluorescence increases and decreases, with cells fluorescing similarly after each 'ON' cycle (Supplementary Fig. 12c). Cells showed significant fluorescence degradation upon switching of Fcn(O) to Fcn(R) in < 45 min of exposure (Supplementary Fig. 12d).

Thus, Fcn (O) amount defines whether protein production increases (high Pyo- and SoxR-mediated induction from P*soxS* promoter, and total protein increase despite ssRA-mediated protein degradation) or decreases (low induction from P*soxS* promoter, and total protein decrease due to degradation). We predicted that by electronically controlling the Fcn (O/R) redox form we could similarly specify increases or decreases in protein levels.

We thus introduce the 'OFF' component of the electronic control scheme, where we stop amplifying gene expression and rely on the biological system (ssRA tag) to drop the output signal in a similar manner as in the chemical experiments. Cells are turned 'ON' with electronic oxidation of Fcn(R) to Fcn(O) ( + 0.5 V) and off with electronic reduction of Fcn(O) to Fcn(R) ( − 0.3 V) (Fig. 3a). In Fig. 3b, we show dynamic control of cell fluorescence with repeated 'ON'/'OFF' electronic signals in a continuous culture. We show that the cells remain responsive and the cycling process is reproducible.

In Fig. 3c, we evaluated the 'ON'/'OFF' profile by varying the cycle time. We found that the fluorescence measured at the half cycle time (after an 'ON' signal) increased monotonically with cycle time. After the half cycle measurement, an 'OFF' signal was passed to the cells. The fluorescence then decreased significantly by the termination of the cycle for half-cycle times above ∼ 45 min. One aspect to keep in mind is that the full electrochemical interconversion between Fcn (O) and Fcn(R) takes about 15 min using our current set-up. Therefore, for the shorter cycle time (for example, 15 min), cells remain exposed to Fcn(O) for the majority of the time. This results in continual gene expression. For a 30 min cycle time, the cells are not in the presence of Fcn(O) for at least half the duration, and we see more pronounced effects of degradation. Longer 'ON' states result in greater fluorescence and longer 'OFF' states result in greater degradation. Fully developed 'ON'/'OFF' switching occurs for cycle times near 30–45 min.

We constructed a simple mathematical model (see Methods and Supplementary Software) to delineate phiLOV synthesis from its degradation (Supplementary Note 5, Supplementary Fig. 13). In Fig. 3d, we show the calculated accumulation of phiLOV in the absence of degradation over time, and found a positive linear relationship between charge and synthesis of phiLOV protein, consistent with data in Fig. 2. This is a

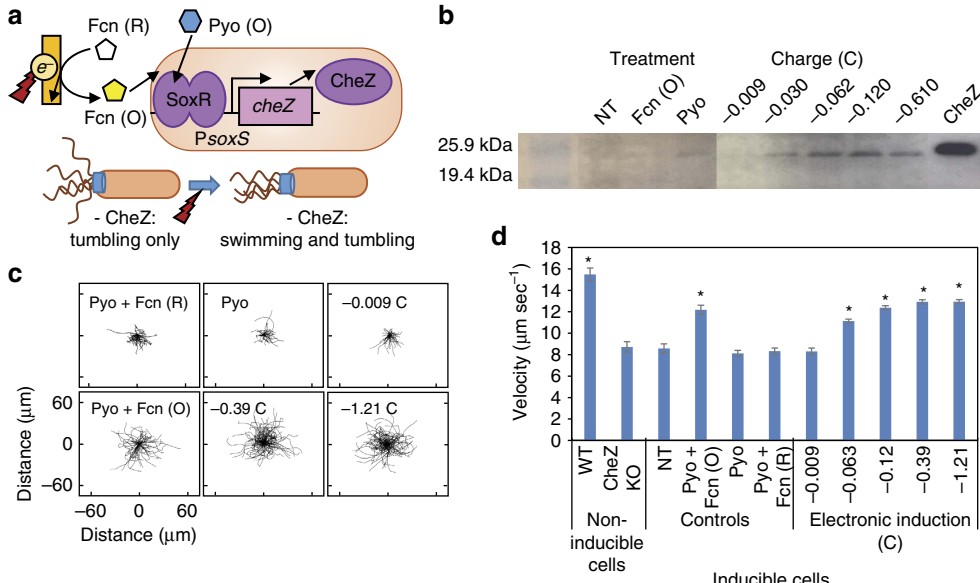

**Figure 4 | Electronic induction of cell motility.** (**a**) Schematic of CheZ induction, which stimulates swimming. *soxR* and P*soxR* omitted from schematic, but present. (**b**) CheZ levels in response to added mediators or electronic induction with Pyo and Fcn (R). NT, no treatment. Samples were processed in parallel. Uncropped blots in Supplementary Fig. 13. (**c**) Two-dimensional recapitulation of 3 s cell trajectories in treated samples as indicated. Samples electronically induced were provided $+0.5$ V with Pyo and Fcn (R) until indicated charge was obtained. (**d**) Cell swimming velocities. Error bars indicate s.e.m. WT are W3110 cells, CheZ KO are isogenic W3110 *cheZ*$^-$, inducible cells are W3110 *cheZ*$^-$ cells transformed with pHW01. *indicates $P < 0.0001$ as analysed by Student's *t*-test against the Pyo + Fcn (R) control (two-tailed). Sample numbers for velocities, starting with WT: 79, 117, 157, 100, 87, 200, 264, 903, 509, 786, 712. In **b,c** and **d** the $-0.009$ C sample indicates 15 min $+0.5$ V application with no mediators.

promising analysis that allows for the prediction of outcomes from the electrogenetic circuit.

**Electronic actuation of bacterial motility.** To show that the redox-driven control scheme can actuate behaviour more complex than fluorescent reporter production, we first electrically induced bacterial swimming. We placed the *E. coli* motility effector gene, *cheZ,* under the P*soxS* promoter, creating pHW01 (details in Methods). CheZ stimulates dephosphorylation of CheY, which drives flagellar motor function and swimming versus tumbling behaviour (Fig. 4a)[45]. CheZ null mutants were transformed with pHW01 and first chemically induced with pyocyanin $+/-$ Fcn(O/R) (Supplementary Note 6). CheZ expression (via western blot) showed similar trends to previously shown fluorescence induction results (Supplementary Fig. 14a). Further, cells stimulated with $+0.5$ V with Pyo and Fcn(R) showed that higher charges correlated with increased CheZ production, almost to the level of wild-type cells (Fig. 4b, Supplementary Fig. 14c shows expanded western blot).

To characterize swimming, we developed a video-analysis algorithm that calculates per-cell swimming velocities (see Methods and Data and Code Availability). CheZ amplification from its background level in the null mutant should correspond to higher velocities as its presence induces more straight swimming and less tumbling. Cell trajectories, showing individual cell paths starting at the origin and spanning 3 s, are smoother and longer with Pyo + Fcn (O) and at higher charges (Fig. 4c). Figure 4d shows that velocity of tracked cells significantly increased with charge. Importantly, we observed no interference on cell motility or CheZ production from non-inducing controls (Supplementary Fig. 15). These results indicate that our redox-mediated approach can electronically stimulate a complex cell behaviour—bacterial swimming—

through gene induction, and do so without apparent interference with motility mechanisms.

**Electronic actuation of bacterial communication.** We aimed to create a bio-electronic cellular information relay: electronically induced cells produce a signalling molecule that is interpreted by a second set of cells that, in turn, responds with altered behaviour specifically encoded by the molecular signal. In this way, we can separate the redox-based electronic-actuation components (relay cell) from the resultant behavioural changes (receiver cell). This could be useful in cases where interactions between Pyo, Fcn(O/R) and the engineered electrogenetic circuit are of background importance. As seen in Fig. 5a, in our relay cell, SoxR induces *Vibrio fischerii* LuxI (instead of phiLOV) expression from the plasmid pTT05. LuxI produces an acylated homoserine lactone (AHL), a bacterial signalling molecule that can diffuse through the membrane to guide quorum sensing (QS) behaviour. The *V. fischerii* LuxI QS system has been widely used to engineer communication networks between non-communicating bacteria[46]. The AHL receiver cell interprets the AHL cue by binding the LuxR protein and expressing phiLOV from the P*luxI* promoter in the plasmid pTT06. As before, adding various Fcn (O) concentrations with Pyo in solution resulted in amplified gene expression in co-cultures of the relay and receiver cells (Supplementary Fig. 16a). Supplementary Fig. 16b shows electronic induction of cell fluorescence of co-cultures over time, the average of which correlates with the charge (Fig. 5b).

In electronically induced co-cultures, the AHL receiver cells exhibited an increase in fluorescence and emerged as a distinct fluorescent population, as can be seen from the flow cytometry histograms in Fig. 5c. Supplementary Fig. 16c shows results of electronic induction between non-co-cultured cells, also with a charge-dependent response (Supplementary Note 7). In addition, quantitative PCR analysis corroborates gene

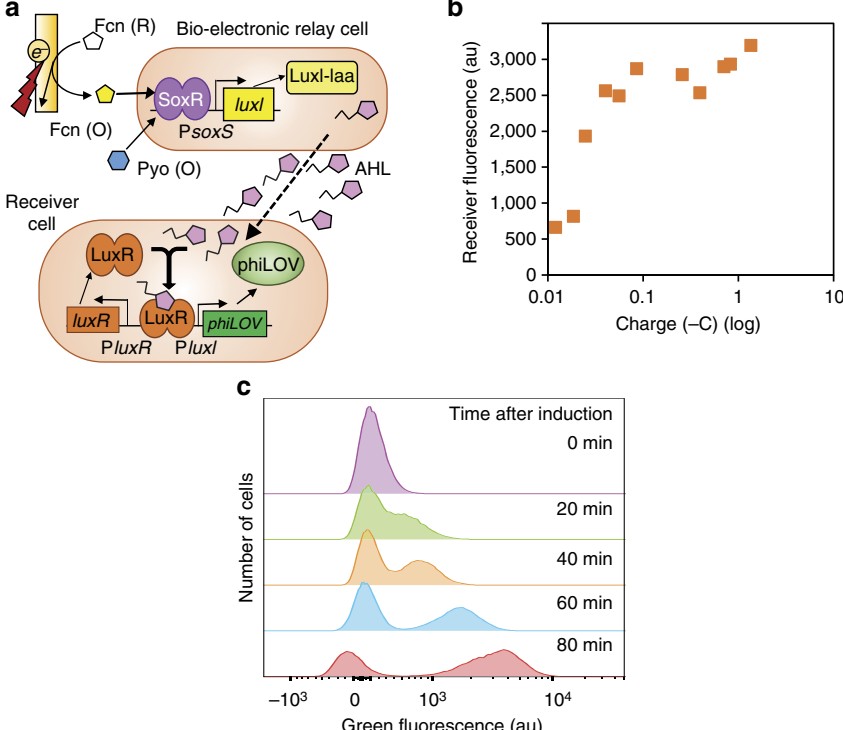

**Figure 5 | Electronic control of cell-to-cell communication.** (**a**) Schematic of electronic control of cell-to-cell communication. Electronic signals modulating Pyo and Fcn (R) to Fcn (O) result in LuxI-Iaa and AHL production from relay cells. *soxR* and P*soxR* omitted from schematic, but present in relay cells. The receiver cells produce LuxR. When LuxR detects AHL, *phiLOV* is induced from the *luxI* promoter. (**b**) Average fluorescence of biosensor cells within co-cultures in which relay cells are electronically induced with the indicated charges. (**c**) Flow cytometry histograms showing the emergence of a fluorescent receiver-cell population at the indicated time points after induction.

expression results for all electronically induced proteins presented (Supplementary Fig. 17), demonstrating messenger RNA decay following the 'OFF' transition in dynamic studies.

These results demonstrate successful biomolecular information transfer through redox-mediated electronic signals to native quorum sensing signalling molecules.

## Discussion

Our work shows, for the first time, the utility of using biologically relevant redox molecules in translating electronic signals to changes in engineered bacterial gene expression. Our system is based on coupling Pyo-driven SoxR activation[31,35] with electronic control of Fcn(O/R) redox form[18,41,42]. This integration allows us to open a new communication pathway and develop a novel framework to connect electronic signals to gene expression. We present in this paper robust evidence and thorough characterization of a functioning bacterial electrogenetic device. To our knowledge, our work is first in demonstrating and characterizing an electrode-based system for reversible and specific redox-driven genetic control in bacteria.

Our applications of this system to genetically induce bacterial motility and cell-to-cell communication highlight its versatility in that it builds upon advances in using electronic control of behaviours that are naturally redox-dependent. Additionally, although we highlight the dynamic gene-actuation capabilities of our system, our aims differ from those of other synthetic biology efforts that enlist non-native components to recognize alternative input signals for precise genetic control using light[47–50] or magnetic and radio waves[51–53]. Instead, we minimally rewire the cells to take advantage of native redox interactions, and

provide insights into their developing role as mediators for bio-electronic communication.

We foresee that our system can be tailored to produce a variety of responses, guide various behaviours, and further the use of other electronic[28,31] and redox-based systems to access and affect biomolecular information transfer, perhaps as part of MFC or BES systems for gene expression based on potential, current or electron acceptor availability. We show preliminary results that expand on the redox mediators, cell genetic background and oxygen levels that are used. Our approach may prove useful for spatio-temporal control of cells immobilized at or near electrode surfaces, for metabolic engineering applications, gut-on-a-chip systems, and other bio-hybrid devices where precise cellular spatio-temporal control is desirable. Additionally, our system offers an additional mode (in addition to light, magnetic, and radio) of relaying electronic signals to cells. Such cells can be programmed to respond to an unprecedentedly wide array of biological and non-biological information and make 'smarter' decisions than previously possible. Our electrogenetic device additionally offers electronic interrogation of SoxRS-specific targets and electron-flow-dependent processes. In sum, our work to translate electronic signals to bacterial gene expression represents a new way of using redox molecules and electron flow for guiding biological function.

## Methods

**Cell strains and plasmids.** The majority of the experiments used *E. coli* DJ901 (*ΔlacU169 rpsL ΔsoxRS901*) (ref. 36). For experiments with CheZ induction W3110 *E.coli* with CheZ genomic deletion were constructed (CheZ KO). Plasmid vectors include pBR322 (for phiLOV and LuxI expression) and pFZY1 (for CheZ expression). Briefly, the complete DNA region encompassing the

soxR (Gene ID: 948566), and the PsoxR and PsoxS promoters (entire region between soxR and soxS) was PCR-amplified from the genome of E.coli MG1655. The genes coding for the proteins phiLOV (fluorescence), LuxI (autoinducer production), or CheZ (motility), with and without ssRA degradation tags, were placed downstream of the PsoxS promoter. Standard restriction cloning techniques and Gibson assembly were used. NEB5α (New England Biolabs, Ipswich, MA) and Top10 Chemically Competent (ThermoFisher Scientific) cells were used for construct assembly. Details of plasmid construction and all sequences are in the Methods and Supplementary Tables 1–3.

**Cell culture.** Cells were grown overnight in lysogeny broth (LB) at 30 °C aerobically with 250 r.p.m. shaking, were inoculated from the overnight cultures at 1.5% in LB, and grown in 37 °C with 250 r.p.m. shaking until $OD_{600}$ 0.2–0.5. The cells were re-suspended in M9 media ($1 \times$ M9 salts, 0.4% glucose, 0.2% casamino acids, 2 mM $MgSO_4$, 0.1 mM $CaCl_2$ and 100 mM MOPS) and then grown at 37 °C in a mini-incubator inside the Coy chamber for anaerobic experiments or in a shaking incubator (250 r.p.m.) aerobically.

**Establishment of anaerobic conditions.** A Coy Laboratory Products (Grass Lake, MI) anaerobic chamber maintained anaerobic conditions—set-up as per the manufacturer's instructions, with nitrogen and $CO_2/H_2/N_2$ mix.

**Spectrophotometric readings.** A SpectraMax M2 plate reader (Molecular Devices, Sunnyvale, CA) was used to read absorbance of ferricyanide (420 nm) and cell amounts (600 nm).

**Cell fixing.** Typically, 100 μl of cells were taken per sample for fluorescence measurements. Cells were washed in PBS, re-suspended in 2% paraformaldehyde in PBS, and incubated for at least 30 min at room temperature before flow cytometry measurements.

**Flow cytometry.** Flow cytometry was performed using a BD Biosciences (Franklin Lakes, NJ) FACS Canto with the BD FACSDiva software. 50,000 cells were collected for each sample and consistently by forward scatter and side scatter. The mean green fluorescence levels of phiLOV (488 nm laser and 530/30 green filter) are based on the means of 40,000–50,000 cells from the number of indicated samples. Analysis was done in FACSDiva, FlowJo and Excel.

**Electrochemical set-up.** For bulk electrolysis, 50 cm-long gold electrodes (0.5 mm diameter, 99.95% metal basis) were wound and used for both working and counter electrodes. An Ag/AgCl reference electrode was used. A CH Instruments, Inc. (Austin, TX) 600-series potentiostat was used for all electrochemical experiments.

Agar salt bridges consisted of 6 inch-long 1.2 mm OD, 0.9 mm ID glass capillary tubes bent into a U shape after brief heating under a Bunsen burner. A 3% agar solution with 1 M KCl was heated and added into the bent capillary tube. Tubes were cooled by immersion in a 3 M KCl solution and stored in 3 M KCl at 4 °C.

Typical electrochemical set-up for Fcn(O/R) interconversion and in situ experiments were performed as follows: the working and reference electrodes were placed in one glass vial with 3 ml of solution and/or cells; in a separate similar vial the counter electrode was placed with another 3 ml of solution or cells. Mediators were added to the counter chamber as follows: if Fcn (O) was added to the working, then Fcn (R) was added to the counter chamber, and vice versa. If pyocyanin was added to the working chamber, then it was also added to the counter chamber. If neither pyocyanin nor Fcn (O/R) were added to the working chamber, then these were also omitted from the counter chamber. Holes to fit the electrodes and salt bridges were punched out in the plastic vial stoppers. Two salt bridges linked the two chambers. A mini magnetic stirrer with a 7 mm stir bar was used to facilitate mixing and accelerate electrochemical conversion in both vials. Unless otherwise stated, oxidation indicates a constant application of $+0.5$ V and reduction $-0.3$ V. For details and picture of the set-up, see Supplementary Fig. 9.

For cyclic voltammograms, scan rates of 0.05 V s$^{-1}$ were used.

***In situ* electronic cell induction.** Cells were cultured as above and placed in the anaerobic chamber. An electrochemical set-up as described above was used—with two chambers, three electrodes and agar salt bridges. Cells at $OD_{600}$ 0.2 (unless otherwise stated) were added to the working electrode vial and placed in the 37 °C mini incubator for ~5 min in order to warm before the addition of mediators.

To initiate the electrochemical signalling, mediators were added, and the working electrode was biased at the indicated voltage for the indicated amount of time. For fluorescent cell sampling, about 100 μl of cells were removed from the solution and fixed as above. If multiple time points were to be taken without further electrochemical signalling, a volume equivalent to 100 μl × number of samples + 100 μl was removed from the glass vial and put in an Eppendorf tube in

the mini-incubator, from which samples were collected. If further electronic signals were to be applied, 100 μl of media + mediators were added back into the culture after sampling. Charge was recorded by the CHI software and the end-point total was used in the figures.

For induction by varying potential, the indicated potentials (Fig. 2b) were applied for 15 min, after which the cells were removed as mentioned above, and sampled every 30 min for 3 h. For induction by varying time, $+0.5$ V was applied for between 10 and 900 s.

Turning cells 'ON' and then 'OFF' involved first cell induction with pyocyanin and Fcn (R) with $+0.5$ V for 15 min. Afterwards, cells were left in the glass vials for the indicated amount of time to produce fluorescence (Fig. 2d indicates total time of induction + culture), and were sampled for fluorescence. To subsequently turn cells 'OFF', an $-0.3$ V reducing potential was applied for 15 min, reduced the Fcn (O), and cells were placed in a separate tube for the remaining time for sampling. Multiple cycles of ON and OFF as in Fig. 3b repeated the above process while cells stayed in the vials with electrodes throughout and fresh media was added when samples were removed.

**Induction of motility.** Overnight cultures were grown as above. Following re-inoculation in LB, cells were grown to an $OD_{600}$ of 0.45 at 37 °C shaking at 250 r.p.m. aerobically. Cells were spun at 400 g for 5 min and re-suspended in an equal volume of M9 media. Cells were placed in the anaerobic chamber where mediators were added as indicated. Non-electrically stimulated samples were induced in the anaerobic chamber at 37 °C for 90 min. Electrically stimulated cells were induced with various charges (constant potential, varying time, as above), after which cells were placed into Eppendorf tubes and cultured for a total of 90 min before analysis. Western Blot analysis is described in Methods and was done using standard techniques. For video analysis, cells were spun down at 400 g for 5 min and re-suspended in chemotaxis buffer (CB: 1 × PBS, 0.1 mM EDTA, 0.01 mM L-methionine, 10 mM D,L-lactate) while still in the anaerobic chamber.

**Motility video analysis.** Cells in chemotaxis buffer were removed from the anaerobic chamber, placed on a microscope slide, and a video was recorded using CellSens software and DX60 microscope equipped with a DP72 camera (Olympus, Waltham, MA). Approximately 100 frames are recorded for each video, using a 20 × objective lens with a green fluorescent protein (GFP) filter.

Motility video analysis was done using Matlab based on methods in literature[54]. Using Otsu's method[55], each frame of the motility video was segmented into a binary image. The built-in function *regionprops* provided the location and shape of each cell. The tracking algorithm uses a nearest-neighbour approach that links cells in subsequent frames based on closeness, size similarity and pixel intensity. The velocity was determined from centroid data. The program accounts for cells that are stuck for part or the entire duration of the video and cells that are under the influence of background flow. In order to create the trajectory diagrams in Fig. 4c, the first 3 s of each cell trajectory in the video are shown, translated and plotted at the origin (0,0).

**Induction of cell-to-cell communication.** The bioelectronic relay cells (DJ901 with the plasmid pTT05) and the receiver cells (DJ901 with the plasmid pTT06) were inoculated from overnight cultures at 1.5% in LB and grown in 37 °C and 250 r.p.m. shaking until reaching $OD_{600}$ 0. 2 − 0.5 aerobically. The cells were re-suspended in the M9 media at an $OD_{600}$ of 0.25 and mixed at a 1:1 relay to receiver cell ratio before induction. Solution-based induction was done as for cells with induced motility above. Electrochemical induction was done as above with application of $+0.5$ V for various times.

**Construction of pTT01-pTT04 plasmids.** The DNA region containing the soxR gene and the region between soxR and soxS was amplified from the E.coli MG1655 genome and ligated into the PCR-Blunt II-TOPO plasmid. The fragment was then digested out with the BamHI and HindIII enzymes and ligated into a similarly digested pBR322 vector. The gene coding for the phiLOV2.1 protein was produced as a gBlock by IDT, with E.coli codons optimized using GenScript from amino-acid sequence from Christie et al.[43] The pTT01 (phiLOV) and pTT02 (phiLOV-LAA) plasmids were assembled using the Gibson Assembly method[56] (NEB Gibson Assembly Master Mix) by PCR amplifying both the phiLOV sequence (with or without the AANDENYALAA (LAA) degradation tag) and the pBR322-soxR-PsoxS constructs with overlaps. The AANDENYADAS (DAS) tag was added to phiLOV by PCR amplifying pTT02, treating the PCR with T4 polynucleotide kinase and ligating with T4 ligase to create plasmid pTT03. The plasmid pTT04 was created by PCR-amplifying pTT03 without the soxR coding sequence, treating the PCR with T4 polynucleotide kinase and ligating with T4 ligase. The relevant primers can be found in Supplementary Table 2. The relevant genetic element sequences, including the tags, can be found in Supplementary Table 3.

**Construction of pTT05 and pTT06.** The plasmid pTT05 was created from pTT01 by PCR amplification without phiLOV. The luxI gene with the LAA tag was amplified from the plasmid pLuxRI2 (ref. 57). Gibson Assembly Master Mix

from NEB was used to assemble the final construct. Plasmid pTT06 was created by amplifying pTT01 plasmid without the *soxR* through P*soxS* region, and *luxR* through *luxI* (including promoters) out of plasmid pTD103Aiia[46]. The Gibson assembly method was used as above. The relevant primers can be found in Supplementary Table 2. The relevant genetic element sequences, including the tags, can be found in Supplementary Table 3.

**Construction of motility plasmid pHW01.** To create the plasmid pHW01 with the *cheZ* gene under control of the P*soxS* promoter, *E. coli* W3110 cells were used as the template for amplifying the *cheZ* and *soxR*-P*soxS* fragments via PCR. The primer set BamHI-SoxR-F and SoxS-cheZ-R was used for the amplification of the *soxR*-P*soxS* fragment, while the primer set SoxS-cheZ-F and CheZ-HindIII-R was utilized for the *cheZ* fragment. In between both PCRs, the primer SoxS-cheZ-R, by our design, was a reverse complementary strand to SoxS-cheZ-F and, therefore, both resulting PCR products shared an overlapping fragment. After gel-extraction of PCR products, both were mixed together at equimolar ratio and an extra PCR was performed with the primers BamHI- SoxR-F and CheZ- HindIII-R for ligating *soxR*-P*soxS*-*cheZ*. The resulting product was inserted into pFZY1 (ref. 58) vector through BamHI and HindIII restriction enzyme cloning.

**Construction of constitutive fluorescent plasmid pT5G.** The plasmid pT5G was derived from a plasmid previously used for constitutive expression of DSRedEx-press2 (refs 59,60). First, a redundant HindIII restriction endonuclease site (AAGCTT) was deleted from plasmid pT5RT7G through plasmid PCR using primers HindIIIdel-F and HindIIIdel-R. The product was phosphorylated with T4 PNK and re-ligated with T4 ligase. Next, the reporter gene *eGFP* was amplified using the t5EGFP-F and t5EGFP-R primers. The *dsRedExpress2* was excited from the pT5RT7G derivative via EcoRI and HindIII digestion and *eGFP* was inserted. Transformation and recovery of the ligation product yielded Top10 + pT5G cells that constitutively expressed EGFP and thereby fluoresced green. The plasmid was transformed into *cheZ* KO cells (below) with the motility plasmid pHW01 to allow for fluorescent video recording.

**Construction of plasmid pTG1.** The *E. coli* K-12 genomic region that constitutes the soxR protein and the divergent overlapping P*soxR* and P*soxS* promoters was inserted into a pCR-BluntII-TOPO plasmid (Thermo Fisher Scientific). This construct and the plasmid pFZY1 were digested with BamHI and HindIII and ligated such that the lacZ gene in pFZY1 was downstream of the P*soxS* promoter. pTG1 allowed for SoxR-mediated expression of β-galactosidase.

**Genomic *cheZ* deletion.** Chromosomal deletion of *cheZ* in *E. coli* W3110 was carried out using the one-step inactivation method described by Datsenko and Wanner[61] In this method, a phage λ Red recombination system was introduced to facilitate the homologous replacement of W3110 *cheZ* gene with kanamycin resistance gene cassette followed by the excision of the resistance cassette for creating *cheZ* knockout of W3110 (W3110 *cheZ*-). Specifically, the Red helper plasmid, pKD46 (GenBank Accession: AY048746.1), was first transformed into W3110 electro-competent cells by electroporation. The transformed cells were grown and selected on LB-agar plates which contained 50 µg ml$^{-1}$ ampicillin at 30 °C. A positive colony was picked and inoculated into 50 ml LB medium which contained 50 µg ml$^{-1}$ ampicillin and 1 mM L-arabinose. The cells were cultivated at 30 °C 250 r.p.m. shaking to an OD600 ~0.3 and electro-competent cells were freshly prepared and kept on ice until the next transformation of a *kan* resistance cassette. To synthesize the kanamycin resistance cassette, we conducted a PCR using the primer set (cheZ-KO-P1F and cheZ-KO-P2R) and the plasmid pKD4 (GenBank Accession: AY048743.1) as the template.

The resulting PCR product of the kanamycin resistance cassette flanked by FLP recognition target sites was produced and gel-purified for subsequent transformation into pKD46 carrying W3110 electro-competent cells above-mentioned. In all, 300 − 500 ng of the kanamycin cassette product was introduced into 50 µl of competent cells by electroporation followed by the incubation with 500 µl SOC medium and 1 mM L-arabinose at 37 °C 250 r.p.m. shaking for 2 h. Cells were grown overnight on an LB-agar plate containing 30 µg ml$^{-1}$ Kanamycin for screening the recombinants. We further isolated colonies from the kanamycin plate and conducted PCR verification for *cheZ* deletion (cheZ_seq-P1F and cheZ_seq-P2R) and kanamycin cassette insertion (primer set 1: cheZ-upstream and k; primer set 2: k2 and cheZ- downstream). Isolated cells were also inoculated in LB medium supplemented with 50 µg ml$^{-1}$ ampicillin for checking the curing of pKD46 plasmid. Subsequently, the removal of the kanamycin resistance cassette from the isolated clones was also implemented by the electro-transformation and temperature upshift induction of the 707-FLPe plasmid. Upon temperature shifting, 30 °C to 37 °C, *cheZ* mutant cells carrying 707-FLPe plasmid expressed FLPe recombinase and then triggered FLP-mediated excision of the FRT-flanked kanamycin resistance cassette. After incubating at 37 °C for 3–5 h, the cells were plated and grown on LB-agar plates. We then picked single clones from the plates and inoculated into LB only, LB with 30 µg ml$^{-1}$ kanamycin, and LB with 3 µg ml$^{-1}$ tetracycline for the screening of kanamycin removal and 707-FLPe plasmid curing. PCR verification of kanamycin cassette removal was further performed with the primer set k1 and kt.

**General cloning procedures.** DNA was extracted from cells using either a Qiagen (Hilden, Germany) or a Zymo Research (Irvine, CA) Miniprep kit according to the manufacturer's instructions. Polymerase chain reaction (PCR) was used to amplify genes or DNA of interest using Q5 DNA Polymerase (NEB). Primers were ordered from Integrated DNA Technologies (IDT, Coralville, IA). NEB restriction enzymes such as BamHI and HindIII were used to generate restriction digests of desired PCR products or plasmids. Agarose gel electrophoresis was used to separate DNA fragments based on size and the gel bands (as visualized with SYBR Safe, Invitrogen) as well as DNA sequencing by Genewiz was used to verify the constructs. Digested fragments were ligated using either NEB Quick Ligase or NEB T4 Ligase. Gibson Assembly was performed with NEB's Gibson Master Mix according to the manufacturer's instructions. Electro- or chemically competent cells (either from NEB, Invitrogen (Carlsbad, CA), electrocompetent, or made with Zymo Research's Z- Competent *E. coli* Transformation Kit) were used for transformation.

**β-galactosidase (Miller) assay.** The Miller assay was performed on ZK126 cells with the pTG1 plasmid expressing β- galactosidase according to standard proto-cols. Miller assay was performed according to standard protocols[62]. Briefly, cells were lysed with chloroform and sodium dodecyl sulfate (SDS) to release β-gal. The substrate ortho-nitrophenyl-β-galactoside was added and cleaved by β-gal into a yellow molecule, *o*- nitrophenol. Absorbance at 600, 550 and 420 nm was quantified by a SpectraMax M2 plate reader. The OD at 600 nm was measured from 250 µl of cells and the ODs at 420 and 550 nm were measured from 200 µl of cells.

**Electrochemical ferricyanide reduction measurement.** To measure ferricyanide reduction by cells, a three electrode set-up was used: an Au working electrode (2 mm diameter, CH Instruments, Inc., Austin, TX), a 4 cm-long platinum wire counter electrode (Alfa Aesar, Haverhill, MA) and Ag/AgCl reference electrode (BASi, West Lafayette, IN). We used about 1.5 ml of cells at OD$_{600}$ of 1.5, with a mini magnetic stir bar (see bulk electrolysis set-up in Methods). The cells were grown as above and incubated inside the anaerobic chamber at 37 °C during measurements. An oxidation potential of + 0.5 V was applied over time to measure ferrocyanide.

**Propidium iodide staining.** Propidium iodide was used to stain dead bacteria. Cells were washed in 10 mM MgSO$_4$ (pH 6.5), then PBS, and finally re-suspended in PBS with 5 µg ml$^{-1}$ of propidium iodide added. The cells were incubated at room temperature while covered with foil for 30 min. Afterwards cells were spun down and re-suspended in PBS. Fluorescence of cells was measured with flow cytometry as described in Methods, with the excitation and emission set for DsRed detection.

**Glucose and acetate measurement.** Glucose was determined by the YSI 2700 SELECT Biochemistry Analyser (YSI Life Sciences, Yellow Springs, Ohio). Acetate was determined by high-performance liquid chromatography, Hewlett Packard 1100 Series using an Aminex resin-based HPX-87H column (Bio-Rad, Hercules, CA). The analysis conditions were as follows: wavelength 210 nm, mobile phase 0.008 N H$_2$SO$_4$, flow rate 0.6 ml per min, temp 35 °C, calibration was done using organic acid analysis standard (Bio-Rad, Hercules, CA).

**qPCR analysis.** To study the messenger RNA levels in response to mediator treatments qPCR was performed. Cells were grown as stated in the Methods, taken to the anaerobic chamber, and let sit for 15 min before treatments. Cells were induced with the indicated mediators for 30 min (if in solution). Electrochemical induction was performed as in the Methods—in all cases + 0.5 V was applied for 15 min, resulting in the indicated charges, after which the cells were cultured a further 15 min before addition of RNA later. Cells were treated as indicated and ~2 × 10$^8$ total cells were washed in equal volume of PBS and then re-suspended and stored in RNA later (Ambion, Austin, TX) at 4 °C overnight. Before RNA isolation, cells were pelleted to remove RNA later. RNA was isolated using the TRIzol Max Bacterial RNA Isolation Kit (Ambion, Austin, TX) according to the manufacturer's protocol, followed by treatment of 50 ng of total RNA with DNase I (Sigma, St Louis, MO). qPCR was performed using SensiFAST SYBR Hi-ROX One-Step Kit (Bioline, Taunton, MA) with ~5 ng of total RNA per reaction using the primers in Supplementary Table 4. Each sample was performed in triplicate (technical replicate). Outlying data was removed. In all, 16s ribosomal RNA was used as the endogenous housekeeping gene. Data was analysed using the ΔΔCt method, with the Ct threshold set automatically by the Applied Biosystems 7300 Real-Time PCR System for all samples.

**Cell preparation for western blotting.** Cells were grown and induced or treated as indicated. For cell lysate preparation, 3 ml of culture were spun down at designated time-points at 6,000 r.p.m. for 5 min. The supernatant was discarded, and the remaining pellets were frozen at − 80 °C. Upon thawing, samples were lysed in 250 µl BugBuster HT (Novagen, Madison, WI) according to the

manufacturer's protocol. Lysate concentrations were assessed via BCA assay (Pierce, Rockford, IL) according to the manufacturer's protocol. Lysates were normalized to $500\,ng\,\mu l^{-1}$ with water and boiled with SDS loading dye.

**SDS–polyacrylamide gel electrophoresis and western blotting.** Purified CheZ was shipped to New England Peptide (Gardner, MA) for antibody generation in rabbit using the Customer Supplied Antigen package. The antiserum was used at a 1:10,000 dilution in a solution of TBST with 25% v/v of cell lysate from $\Delta cheZ$ E. coli, as indicated in the below.

The secondary horseradish peroxidase-conjugated secondary antibody (Catalog number: 65-6120, Invitrogen, Carlsbad, CA) was diluted as indicated in the below.

Approximately 12 μg total protein per sample was loaded in a 12.5% SDS–polyacrylamide gel electrophoresis gel and run at 120 V in running buffer (25 mM Tris, 192 mM glycine, 0.1% SDS, pH8.3). The proteins were then transferred to a nitrocellulose membrane in transfer buffer (48 mM Tris, 39 mM glycine, 20% methanol, pH 9.2) using the semi-dry Trans-Blot SD cell (Bio-Rad, Hercules, CA). Blots were blocked overnight at 4 °C with Tris-buffered saline (20 mM Tris, 500 mM NaCl, pH 7.5) with 0.1% Tween-20 (TBST) and 10% nonfat milk. TBST with 3% bovine serum albumin and a 1:10,000 dilution of anti-CheZ rabbit antiserum is incubated for at least 30 min, shaking at room temperature, with 25% total volume of cell lysate from W3110 $cheZ^-$ cells.

W3110 $cheZ^-$ lysate is prepared by growing a volume (50 ml) of the cells overnight, pelleting the next day, re-suspending in 40% the volume (20 ml) of TBST with 100 μl Triton X-100 (Bio-Rad, Hercules, CA), sonicating for 30 min or until lysate is coloured and remaining pellet is small. After rinsing the membrane in TBST, the blot is incubated with the primary antibody mixture for 90 min. The membrane is thoroughly rinsed again, and incubated for 60 min with an horseradish peroxidase-conjugated secondary antibody (Sigma, St Louis, MO) diluted 1:4,000 in TBST with 3% bovine serum albumin. The blot was imaged using a chemiluminescence detection system (ECL; Pierce, Rockford, IL) according to the manufacturer's instructions, and developed using Hyperfilm (GE Healthcare, Waukesha, WI). Supplementary Fig. 14 shows molecular size markers and uncropped blots.

**Data availability.** The data that support the findings of this study are available from the corresponding author upon request. The Matlab code used for velocity analysis is available at http://bentley.umd.edu in the 'Selected Publications' section under Pottash et al.[63] The code for the model of synthesis and degradation is available in the Supplementary Software.

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

## Acknowledgements

We would like to acknowledge funding provided for this work by the Defense Threat Reduction Agency (DTRA, HDTRA1-13-1-00037) and the National Science Foundation (CBET 1160005 and CBET 1264509), and the NSF EAPSI fellowship to T. (Gordonov) Tschirhart. We would like to thank Dr James A. Imlay for helpful conversations.

## Author contributions

T.T., E.K., G.F.P. and W.E.B. developed the concepts. T.T., H.-C.W. and A.Z. genetically engineered the constructs. H.U. and A.E.P developed and ran Matlab tools and analyses. T.T., R.M. and A.N. designed and ran the experiments. T.T., E.K., R.M., H.U., G.F.P. and W.E.B. analysed the data. T.T., G.F.P. and W.E.B. wrote and edited the manuscript and figures. J.S., G.F.P. and W.E.B. supervised the work.

## Additional information

**Competing financial interests:** The authors declare no competing financial interests.

