## [Peer Review File · Nature Communications]

Reviewers' Comments:

Reviewer #1 (Remarks to the Author)

NCOMMS-16-07646 - Electronic control of bacterial gene expression and cell behavior."

The authors show that electronic information can be used to control bacterial cell behavior - gene expression, motility and inter-cellular communication. Combining standard electrotransfer compounds and the well-characterized native SoxR - SoxR target promoter, they managed to reversibly and dose-dependently control target gene expression by an electrode in a completely anaerobic condition. This electrogenetic interface capitalized on the alteration of the redox state of SoxR.

Major concerns

1. The authors test their device in a strictly anaerobic environment, which will likely limit this technology in real-world applications.
2. As proof of concept the authors link their electrogenetic device to expression of state-of-the-art chemotaxis and quorum sensing. Since these chapters neither contain any innovative information nor provide any advancement, they distract from the major focus of designing a bacterial electrogenetic interface. Enthusiasm would dramatically increase if the authors would provide a compelling real-world application that would appeal for a broader audience. Since the assembly of this device was rather simple and straightforward, showing that the device is useful for a real-world application should not be beyond the scope of the study, but would be highly appreciated by the interdisciplinary readership of N. Commun.
3. The authors use E. coli DJ901, which they declare as exclusive SoxR deletion mutant. However, to my knowledge E. coli DJ901 is $\Delta(\text{soxS-soxR})$, and therefore lacks not only SoxR but also the SoxS regulon consisting of some 40 genes. The genetic background of the E. coli should be clarified, in particular, since the authors mention soxS on several occasions. It remains elusive whether they refer only to the soxS promoter and/or the soxS protein, which may be a transcription modulator itself and so interfere with the electrogenetic device. Also, E. coli DJ901 is complemented with an episomal copy of SoxR, either expressed from a constitutive promoter (pTT01) or from the divergent overlapping soxRS promoters (pTT02, pTT03, pTT05). Since SoxR represses both of the overlapping soxRS promoters but activates the soxS promoter the genotype of the E. coli and the electrogenetic device needs to be clarified and presented in clear detail throughout the manuscript; including the figures.
4. The electrogenetic device is not characterized in sufficient detail. Most of the activity and dynamics of the electrogenetic device is based on SoxS-dependent fluorescent reporter output. Since this is an electrogenetic device, the oxidative state of the SoxR iron cluster playing the key role needs to be characterized by EPR spectroscopic analysis of entire cells (Koo et al., 2003; EMBO J 22:2614). This experiment would demonstrate the direct link between the electrical charge and the oxidation state of SoxR and is barely needed.
5. When characterizing the electronic signal to cycle the gene expression "ON" and "OFF", the authors mention that the system is switched off by reduction of the ferricyanide that leads to "no further electron to amplify the gene expression". However, the state of SoxR remains unclear. SoxR likely remains in an active form (oxidized state). The authors should provide a possible explanation of the SoxR reduction mechanism that halts transcription and experimentally confirm their statement.

6. The qPCR analysis of electrochemically-induced cells is a critical experiment, which should be repeated including a soxR control.

Minor points

1. In their first sentence of the introduction the authors set the stage mentioning "electrocardiograms", "pacemakers" and "interactions of electrodes with the nervous system". Provided that the authors focus on a bacterial electrogenetic device only working under anaerobic conditions - likening their work to therapeutic opportunities seems to be distracting. This is particularly important as electrogenetic devices have indeed already been implemented in mammalian cells under aerobic conditions (Weber et al., 2009; NAR 37: e33). The authors may want to cite this work and revise some of their "first" statements throughout the manuscript.

2. Page 6 "Thus, ferricyanide amount sets a balance between protein production and degradation." This statement needs clarification; does ferricyanide affect the half-life of an ssRA-tagged protein or only act at the transcriptional level?

3. Page 7 "In sum, chemical studies demonstrated that (i) Pyocyanin induces phiLOV gene expression by soxS-mediated regulation." Is it a soxS-dependent regulation or soxS promoter-dependent transcription? As mentioned above the presence or absence of the soxS gene should be clarified in this genetic background.

4. Page 9 "For shorter cycle times, the "OFF" period is shorter relative to the 15 min "ON" window, yielding less available time for attenuation of expression and degradation." What is the reason for using the term "attenuation" in this transcriptional control system? The same applies for degradation, is this effect related to protein or the messenger RNA?

Reviewer #2 (Remarks to the Author)

Summary of the key results:

Tschirhart et al report electrochemical actuation of cellular behavior in Escherichia coli. The authors modulate the redox state of the mediator ferricyanide in the presence of pyocyanin to alter the redox state of the transcriptional activator SoxR. SoxR in turn regulates expression of heterologous genes downstream of the soxS promoter. The authors convincingly demonstrate electronic-driven expression of fluorescent, motility, and cell-signaling proteins that alter cellular behavior. Although their work is not the first example of regulating cellular behavior electronically using redox molecules, their approach of using redox mediators to activate native transcriptional regulators and thus upregulate expression of synthetic gene circuits is novel and potentially high-impact.

Despite these strengths, this manuscript has several crippling shortcomings which greatly diminish its impact. Firstly and most importantly, one of the central conclusions of the paper, that redox interactions control the respiratory electron transfer chain to amplify gene expression, is not sufficiently supported. Secondly, the language throughout the manuscript is imprecise and occasionally incorrect, which confuses the data and interpretation. Thirdly, the supporting information is poorly selected, organized, and referenced throughout the manuscript, leaving the reader floundering in an ocean of data. Addressing these shortcomings adequately will require extensive and significant changes to the claims, content, and organization of the manuscript.

Originality and interest: if not novel, please give references

The claim that this is the first example of transducing electronic information via redox molecules to actuate cellular behavior (pg 3, 2nd paragraph, 1st line) is inaccurate. There are several examples of altering the behavior of metal-reducing bacteria electronically. These examples have been the

subject of recent reviews, e.g. *Biotechnol. Bioeng.* 113, 687-697 (2016), *Appl Microbiol Biotechnol* 98, 509-5 (2014). Also, previous studies have actuated *E. coli* behavior, i.e. substrate utilization, using both redox mediators (e.g. *Bioresource Technology* 195, 57-65 (2015)) and redox proteins (e.g. *ChemElectroChem* 1 1874-1879 (2014)).

Nonetheless, as I understand it, the approach presented in the manuscript offers several significant advantages over these previous efforts; in addition to engineering simplicity and versatility (which the authors discuss), this approach offers specificity, i.e. most significantly impacting SoxR-regulated targets, unlike previous examples using mediators. The authors need to revise the manuscript to better frame their work in the context of previous work so that they can more accurately highlight the originality and advances of their approach.

Data & methodology: validity of approach, quality of data, quality of presentation

The presentation of the data is extremely spotty. The SI material and cell schematics (most critically Figure 1b, but also Fig 4a, 5a) need significant editing. While there are 23 supplemental figures in the SI, many of them are not referred to in the main text. Those figures that are referenced are not referenced in order. And on top of that, many times there is a lot of extraneous data, i.e. the 10 different concentrations of Pyo and Fcn in Supp. Figure 8 when only the concentration used for the main experiments are needed. This same problem crops up when discussing the dynamics of electronically-controlled gene expression. There is an overabundance of data presented about this topic (S Fig 6, S Fig 7, Fig 3), which obfuscates the central point. The key point is that the steady-state protein concentration is well-described by the electrochemically-dependent rate of protein synthesis (linearly dependent on the amount of charge) and the observed protein degradation rate (Fig 3f).

Moreover there is not a clear logical flow in the paper, most particularly the in the first and third subsections of the results Redox mediator effects on cells and engineered gene expression and Electronic signals to cycle gene expression "ON" and "OFF". For example, the results switch between evidence about the mechanism of increased gene expression (pg 5 last paragraph) to the dynamics of gene expression (pg 6 first paragraph), back to the mechanism (pg 6 second paragraph). This poor organization makes it hard for the reader to follow the arguments presented.

Regarding the electrochemical ferricyanide reduction measurements, there is no clear explanation as to why the given applied potentials were chosen. S Fig 14 does not explain this as it is just a representation of the electrochemical setup. In addition, it would be beneficial to incorporate cyclic voltammograms of ferro/ferricyanide in biotic and abiotic samples in the same figure. Further discussion should follow regarding the R and O peaks in both scenarios. Other information such as CV scan rates would be helpful.

Appropriate use of statistics and treatment of uncertainties

The use of statistics was appropriate.

Conclusions: robustness, validity, reliability

The data presented in Figure 1 and SI Figures 11, 12 indicate that addition of a terminal electron acceptor is needed to reach the highest expression of SoxR-regulated genes in the presence of pyocyanin. However, the conclusion that this mechanism driving this amplification of the pyocyanin response is indeed interaction of ferricyanide and pyocyanin with the respiratory ET chain as depicted in Figure 1b is not adequately supported. This mechanism is unsupported because of the following reasons:

The claim that pyocyanin can be re-oxidized by quinone components of the electron transport chain machinery (pg 4) is not supported by the reference indicated. Reference 31 does not discuss pyocyanin.

The claim that experiments under anaerobic conditions kept SoxR activation at low levels, below toxicity, (pg 4) is unsupported by SI Figure 12. SI Figure 12 does not address the level of SoxR activation or the overall cell viability under anaerobic conditions.

If the mechanism is as indicated, then using 5 mM Fcn(R/O) should enable the effective pyocyanin (O) concentration to be 5mM instead of 5 μ M, i.e. a 1000-fold increase. Why there is only a \sim 17 fold increase in phiLOV expression?

The cell growth data presented in Supplementary Figure 8 is very unusual and suggests that growth is being altered under conditions that enable electronic control. Specifically, anaerobic growth of *E. coli* on glucose does not typically show the two phases presented in SFig 8a and the final OD is usually much higher, e.g. PLoS ONE 4(2): e4432. doi:10.1371/journal.pone.0004432. Lastly, SFig 8a indicates that the combination of redox mediators and pTT03 yields a growth defect. This change in growth matters because if cell growth is slowed, then the rate of protein dilution due to cell growth goes down and the steady-state protein concentration goes up. In short, the conditions of electronic control could slow growth and thus lead to increased fluorescence protein levels.

The claim that PMS generates comparable results is not supported by a figure or other data (pg 6).

Under anaerobic conditions, *E. coli* will readily ferment glucose, so electrons do not necessarily have to be entering the respiratory chain under these conditions. To directly implicate aspects of the electron transport chain in the effect seen here, the authors would need to show that quinone-deficient mutants, for example, do not show this same amplification. Also, showing that you get the same effect with a non-fermentable electron donor would also significantly help implicate respiration.

In short, while there data in the manuscript suggests that respiration is involved, the data is not strong enough to exclude alternative hypotheses and conclude that the mechanism is respiration-driven.

Suggested improvements: experiments, data for possible revision

Please see section above.

References: appropriate credit to previous work?

As noted previously, the manuscript does not place the work appropriately in context because it fails to reference key areas of related work. The introduction needs to address the successes and shortcomings of using redox mediators by themselves to actuate cellular behavior, e.g. unbalanced fermentation, electrosynthesis; this is a significant literature, so reviews are helpful. Additionally, there are more sophisticated examples of using redox mediators and synthetic biology to electrochemically actuate behaviors, e.g. biosynthesis, gene expression, and this work needs to be cited.

Clarity and context: lucidity of abstract/summary, appropriateness of abstract, introduction and conclusions

The manuscript lacks clarity because of repeated use of imprecise or incorrect words. For example, the authors refer to the 'gene amplification' multiple times (pg 7 line 6), when they are really referring to an increase in protein concentration, not the number of genes (DNA). On a more substantive issue, the authors describe using a degradation tag to switch gene expression "OFF" (pg 2, paragraph 1). This description is incorrect. A degradation tag does not significantly affect the process of gene expression (DNA->RNA->protein). Rather, it decreases the lifetime of that protein, so that the protein concentration more closely resembles the dynamics of gene expression and is less affected by the rate of cell growth. While the authors eventually address this point (pg 10), the initial discussion has already led the reader astray.

Minor comments:

The title is far too broad. It should specify the organism and the use of redox mediators in

combination with synthetic genetic circuits

The first two paragraphs of Results section seem more suitable for the Introduction section.

The statement that this work opens the doors for engineering cells that make decisions based on electronically relayed information (Discussion), i.e. electromagnetic, is incorrect. The field of optogenetics is all about engineering cells so that their behavior can be regulated by light. The authors cite this work in the introduction, so this statement is quite puzzling.

Reviewer #1 (Remarks to the Author):

NCOMMS-16-07646

The authors show that electronic information can be used to control bacterial cell behavior - gene expression, motility and inter-cellular communication. Combining standard electrotransfer compounds and the well-characterized native SoxR - SoxR target promoter, they managed to reversibly and dose-dependently control target gene expression by an electrode in a completely anaerobic condition. This electrogenetic interface capitalized on the alteration of the redox state of SoxR.

Major concerns

1. The authors test their device in a strictly anaerobic environment, which will likely limit this technology in real-world applications.

We thank the reviewer for this note and believe there are many opportunities beyond those in the absence of oxygen. As we now state in the manuscript, testing our system in anaerobic conditions provides an added degree of control over the redox molecules and processes. Since oxygen interacts with pyocyanin and “competes” with Fcn (O) as an electron acceptor, omitting it from the system allowed us to better study and control the redox interactions. Nevertheless, we believe there are many real-world applications for this work and that these applications span oxygen ranges including strictly anaerobic conditions as well as aerobic systems. For example, much work in our lab is targeted towards understanding the GI tract and the human microbiome. We and others are developing in vitro systems that recapitulate the GI tract. Oxygen gradients are steep; from strictly anaerobic environments in the lumen to fully saturated with oxygen just mm away. Endoscopic devices are envisioned that provide chemical information. They might contain bacterial cells as reporters of molecular information and they could be programmed to deliver molecules to specific regions. Another example is the bacterial fuel cell and similar technologies, where cells might be programmed to carry out specific functions in addition to providing for energy transfer. Finally, we envision many opportunities in metabolic engineering for the production of small molecules where this technology could be used to actuate gene expression in an unconventional manner, perhaps orthogonal to the system of study.

Most importantly, to show our system’s potential in conditions that are not strictly anaerobic, we provide additional data. In Supplementary Figure 5, we show fluorescence induction in DJ901 cells that are cultured outside of an anaerobic chamber – with and without aeration (shaking). Here, the sensitivity to pyocyanin is at far lower concentrations, and we show again that fluorescence increases with Fcn(O) addition. Future work could target the bounds of oxygen limitation/influence, but for now, we are pleased to first report electrogenetic induction.

2. As proof of concept the authors link their electrogenetic device to expression of state-of-the-art chemotaxis and quorum sensing. Since these chapters neither contain any innovative information nor provide any advancement, they distract from the major focus of designing a bacterial electrogenetic interface. Enthusiasm would dramatically increase if the authors would provide a compelling real-world application that would appeal for a broader audience. Since the assembly of this device was rather simple and straightforward, showing that the device is useful for a real-world application should not be beyond the scope of the study, but would be highly appreciated by the interdisciplinary readership of N. Commun.

We appreciate the reviewer’s support. Our goal was to first provide a thorough characterization of a novel electrogenetic device. More complex applications are in development and we feel these would require their own manuscripts. We fear the method of electrogenetically stimulating gene expression would be associated with the application rather than a stand-alone advancement. We believe the focus on electrochemical information transfer to biological systems, where molecules convey information (e.g., gene expression and bacterial signaling) is sufficiently novel and impactful.

3. The authors use E. coli DJ901, which they declare as exclusive SoxR deletion mutant. However, to my knowledge E. coli DJ901 is $\Delta(\text{soxS-soxR})$, and therefore lacks not only SoxR but also the SoxS regulon

consisting of some 40 genes. The genetic background of the *E. coli* should be clarified, in particular, since the authors mention *soxS* on several occasions. It remains elusive whether they refer only to the *soxS* promoter and/or the *soxS* protein, which may be a transcription modulator itself and so interfere with the electrogenetic device. Also, *E. coli* DJ901 is complemented with an episomal copy of SoxR, either expressed from a constitutive promoter (pTT01) or from the divergent overlapping *soxRS* promoters (pTT02, pTT03, pTT05). Since SoxR represses both of the overlapping *soxRS* promoters but activates the *soxS* promoter the genotype of the *E. coli* and the electrogenetic device needs to be clarified and presented in clear detail throughout the manuscript; including the figures.

We clarify by explicitly including the genotype of DJ901; the reviewer is correct in that there are a slew of genes that have been deleted. The reason our method works so well, as the reviewer has implied, is that the cells are generally less able to respond to oxidants like pyocyanin and ferricyanide by eliciting other gene responses and protective proteins. These would potentially cloud the response we've been able to achieve. We further note that the bacterial motility induction studies were performed in the strain W3110 $\Delta cheZ$ which still had its' native *soxRS*. We provide new data in Supplementary Figure 2 from *E. coli* ZK126 ($\Delta(argF-lac)169 \lambda^{-} IN(rrnD-rrnE)1 rph-1 tnaA5$) which is a β -gal null mutant. The system works well here too.

Additionally, we provide data from DJ901's isogenic parent GC4468. We chose to work with DJ901 because of its' better responsiveness, not because it was the only genotype that allowed electroactuation. When we describe *soxS* elements in our plasmid vectors, we meant the promoter regions. We have clarified this by using "P*soxS*" when we refer to the promoter in the revised manuscript. Plasmid maps in Supplementary Figure 3, our description of plasmid construction in the Supplementary Methods, and the information in Supplementary Table 1 all point to our genetic components.

4. The electrogenetic device is not characterized in sufficient detail. Most of the activity and dynamics of the electrogenetic device is based on SoxS-dependent fluorescent reporter output. Since this is an electrogenetic device, the oxidative state of the SoxR iron cluster playing the key role needs to be characterized by EPR spectroscopic analysis of entire cells (Koo et al., 2003; EMBO J 22:2614). This experiment would demonstrate the direct link between the electrical charge and the oxidation state of SoxR and is barely needed.

EPR characterization of SoxR's iron sulfur clusters has previously been performed with redox cycling drugs (work cited by reviewer and Singh et al. Mol Microbiol. 2013 Dec;90(5):983-96). Since we did not see activation of gene expression in our electrochemical-signal-only or Fcn(R/O)-only controls, we did not deem it necessary to delve into the SoxR oxidation state. Our system behaved "as expected" with regards to established SoxR-redox drug interactions. Further experiments would help characterize the mechanism and also help to identify the potentially many components involved. We feel that the system level response is the most important and also suggest that it is actually because of the many years of study of the *SoxRS* regulon that we were even able to attempt this work.

That is, we think it is important to highlight that we have based the entire study on *SoxRS* principally because its regulation and activity are well known. When this is coupled with the literature surrounding electron carriers of the respiratory chain and their interactions with drugs such as ferricyanide, we believe many in the field will build off of this work and develop interesting and useful applications.

5. When characterizing the electronic signal to cycle the gene expression "ON" and "OFF", the authors mention that the system is switched off by reduction of the ferricyanide that leads to "no further electron to amplify the gene expression". However, the state of SoxR remains unclear. SoxR likely remains in an active form (oxidized state). The authors should provide a possible explanation of the SoxR reduction mechanism that halts transcription and experimentally confirm their statement.

The SoxR protein contains iron-sulfur clusters, which are typically in the reduced state and keep SoxR inactive. The oxidation of the clusters activates SoxR and transcription from the *soxS* promoter. To both keep SoxR in the normally-inactive state and to re-reduce it after it is oxidized, NADPH-dependent enzymes have been implicated (Koo, M. S. et al. EMBO J 22, 2614-2622, (2003)). We clarify this in the

manuscript.

6. The qPCR analysis of electrochemically-induced cells is a critical experiment, which should be repeated including a soxR control.

We provide additional qPCR analysis of the fluorescent reporter for cells turned "ON" and "OFF" in Supplementary Figure 17. As noted above, the overall point of the work is to demonstrate protein and even population level responses through the electronic actuation of specific promoters. The promoters selected are well described in the literature, as noted above, and our analysis is simply meant to show that no atypical behavior is seen and to corroborate protein expression results.

Minor points

1. In their first sentence of the introduction the authors set the stage mentioning "electrocardiograms", "pacemakers" and "interactions of electrodes with the nervous system". Provided that the authors focus on a bacterial electrogenetic device only working under anaerobic conditions - likening their work to therapeutic opportunities seems to be distracting. This is particularly important as electrogenetic devices have indeed already been implemented in mammalian cells under aerobic conditions (Weber et al., 2009; NAR 37: e33). The authors may want to cite this work and revise some of their "first" statements throughout the manuscript.

We agree that mentioning "electrocardiograms" etc. distracts from the main point of the paper and have removed these confounding statements. We do cite the work the reviewer mentions (reference 43 in our previous manuscript), but agree that it should be featured more prominently and at the outset (see the 3rd paragraph of the introduction).

Regarding our use of the word "first" – our work is indeed the first electrogenetic device in bacteria, to our knowledge. We also agree that describing our work as the first to control "behavior" electronically can be misleading. By this, we meant behavior (exemplified by cell motility and quorum sensing) that was derived from genes transcribed from the electronically-induced PsoxS promoter. There are several examples in the literature referring to "controlling" behavior electronically, but ours is the first where one can specify control through directed and specific gene expression. Our work is differentiated from others which show altered behavior that is naturally dependent on redox changes or electron movement. We clarify in the 2nd and 3rd paragraphs of the introduction, framing our work more appropriately to the field.

2. Page 6 "Thus, ferricyanide amount sets a balance between protein production and degradation." This statement needs clarification; does ferricyanide affect the half-life of an ssRA-tagged protein or only act at the transcriptional level?

We clarify this statement in the 1st paragraph of the section "Dynamic control of gene expression". The reviewer's comment is well taken, the Fcn (O) amount defines whether protein production increases (high Pyo- and SoxR-mediated transcription from PsoxS promoter, and total protein increase despite ssRA-mediated protein degradation) or decreases (low transcription from PsoxS promoter, and total protein decrease due to degradation).

3. Page 7 "In sum, chemical studies demonstrated that (i) Pyocyanin induces phiLOV gene expression by soxS-mediated regulation."

Is it a soxS-dependent regulation or soxS promoter-dependent transcription? As mentioned above the presence or absence of the soxS gene should be clarified in this genetic background.

We clarify that this is PsoxS promoter dependent transcription.

4. Page 9 "For shorter cycle times, the "OFF" period is shorter relative to the 15 min "ON" window, yielding less available time for attenuation of expression and degradation."

What is the reason for using the term "attenuation" in this transcriptional control system? The same applies for degradation, is this effect related to protein or the messenger RNA?

Since we are measuring protein amounts, the degradation applies to protein degradation (ssRA-

mediated, as mentioned earlier), not mRNA. We use the term “attenuation” to reflect the systems level output; we use the level of Fcn(O)/(R) to tune protein quantity (see answer to #2 above).

Reviewer #2 (Remarks to the Author):

Summary of the key results:

Tschirhart et al report electrochemical actuation of cellular behavior in Escherichia coli. The authors modulate the redox state of the mediator ferricyanide in the presence of pyocyanin to alter the redox state of the transcriptional activator SoxR. SoxR in turn regulates expression of heterologous genes downstream of the soxS promoter. The authors convincingly demonstrate electronic-driven expression of fluorescent, motility, and cell-signaling proteins that alter cellular behavior. Although their work is not the first example of regulating cellular behavior electronically using redox molecules, their approach of using redox mediators to activate native transcriptional regulators and thus upregulate expression of synthetic gene circuits is novel and potentially high-impact.

Despite these strengths, this manuscript has several crippling shortcomings which greatly diminish its impact. Firstly and most importantly, one of the central conclusions of the paper, that redox interactions control the respiratory electron transfer chain to amplify gene expression, is not sufficiently supported.

Reviewer #2 points out that one of our main conclusions, that we elucidated the mechanism of response amplification to be through redox interactions with the electron transport chain, is not substantively supported. We believe there is a misunderstanding here - we do not claim that a mechanistic underpinning of the methodology is demonstrated. There are simply too many factors to begin to peel apart the specific actors of the respiratory chain that are playing a role. We regret that our text seemed to overstep our bounds. Our intent is to make inferences about the mechanism from a rich literature and from the experiments presented in this paper. For example, the fact that nitrate, another terminal electron acceptor, produces similar increases in gene expression as ferricyanide supports our conclusions. This point was buried in the previous manuscript. These inferences helped guide us in our experiments, and at no time did we find evidence to the contrary of our main hypotheses. We fully realize that in order for a detailed mechanistic understanding to be conclusive, much more work will need to be done.

We present in this paper robust evidence and thorough characterization of a functioning electrogenetic device. We provide evidence for initiation as well as amplification of gene expression, the latter by interfering with the respiratory chain. We believe that additional mechanistic studies may provide additional insight and opportunity for enhancement, but we believe these studies are beyond the scope of the current work.

Secondly, the language throughout the manuscript is imprecise and occasionally incorrect, which confuses the data and interpretation.

We acknowledge this and have revised substantially. We address the specific concerns below.

Thirdly, the supporting information is poorly selected, organized, and referenced throughout the manuscript, leaving the reader floundering in an ocean of data. Addressing these shortcomings adequately will require extensive and significant changes to the claims, content, and organization of the manuscript.

We concede that we had volumes of supporting information. We have substantially reduced this data to include only information that directly supports the main conclusions and, further, we have re-organized the supporting data so that appears sequentially with the text; a more conventional style.

Originality and interest: if not novel, please give references

The claim that this is the first example of transducing electronic information via redox molecules to actuate cellular behavior (pg 3, 2nd paragraph, 1st line) is inaccurate. There are several examples of altering the behavior of metal-reducing bacteria electronically. These examples have been the subject of recent reviews, e.g. *Biotechnol. Bioeng.* 113, 687-697 (2016), *Appl Microbiol Biotechnol* 98, 509-5 (2014). Also, previous studies have actuated *E. coli* behavior, i.e. substrate utilization, using both redox mediators (e.g. *Bioresource Technology* 195, 57-65 (2015)) and redox proteins (e.g. *ChemElectroChem* 1 1874-1879 (2014)). Nonetheless, as I understand it, the approach presented in the manuscript offers several significant advantages over these previous efforts; in addition to engineering simplicity and versatility (which the authors discuss), this approach offers specificity, i.e. most significantly impacting SoxR-regulated targets, unlike previous examples using mediators. The authors need to revise the manuscript to better frame their work in the context of previous work so that they can more accurately highlight the originality and advances of their approach.

Reviewer #2 is concerned with our claim being the first example of transducing electronic information via redox molecules to actuate cellular behavior. As R2 points out, there are indeed many examples in altering bacterial behavior via redox molecules that interact with various redox-responsive components in bacteria. Our method enables electronic control of phenotypic behavior that results from specific gene expression. We demonstrate this versatility by electronically controlling quorum sensing and motility behaviors – which are not naturally redox responsive. We suggest this difference is key and makes our method widely applicable. We believe the reviewer's concern stems from our previously "cloudy" text. In our revision, we clearly state what makes our system different and versatile. We also place our work in the context of previous literature.

As mentioned above, it was our error to suggest this is a first example of controlling behavior electronically; we meant behavior resulting from intended and specific gene expression that is electronically actuated. Both Reviewer 2 and Reviewer 1 mentioned this strength and noted that our approach is original in this regard. Our electrogenetic device is indeed simple and versatile. The examples in the abovementioned literature rely on behavior that is naturally dependent on redox interactions. Indeed, our references 12-23 include several of the reviewer's references along with other similar studies. This body of literature largely addresses metabolism, electron movement etc. and various derived phenomena (microbial fuel cell function, substrate production). We believe the revised manuscript is far superior to the previous version.

Data & methodology: validity of approach, quality of data, quality of presentation

The presentation of the data is extremely spotty. The SI material and cell schematics (most critically Figure 1b, but also Fig 4a, 5a) need significant editing. While there are 23 supplemental figures in the SI, many of them are not referred to in the main text. Those figures that are referenced are not referenced in order. And on top of that, many times there is a lot of extraneous data, i.e. the 10 different concentrations of Pyo and Fcn in Supp. Figure 8 when only the concentration used for the main experiments are needed. This same problem crops up when discussing the dynamics of electronically-controlled gene expression. There is an overabundance of data presented about this topic (S Fig 6, S Fig 7, Fig 3), which obfuscates the central point. The key point is that the steady-state protein concentration is well-described by the electrochemically-dependent rate of protein synthesis (linearly dependent on the amount of charge) and the observed protein degradation rate (Fig 3f).

We edited our cell schematics to further clarify our system and have removed parts of figures that may have presented extraneous information. Some of these we included in the revised Supporting information. Also, we have substantially reduced our Supporting Information to include that which more directly supports the main conclusions. Since this is the first study characterizing such an electrogenetic device, we believe showing data wherein we tested several concentrations of effector molecules and their conditions is important. In this way, we establish limits and system responses. Readers interested in seeing the protein levels at various mediator amounts, charges, etc. will be able to do so by looking at this supplementary data.

Moreover there is not a clear logical flow in the paper, most particularly the in the first and third

subsections of the results Redox mediator effects on cells and engineered gene expression and Electronic signals to cycle gene expression "ON" and "OFF". For example, the results switch between evidence about the mechanism of increased gene expression (pg 5 last paragraph) to the dynamics of gene expression (pg 6 first paragraph), back to the mechanism (pg 6 second paragraph). This poor organization makes it hard for the reader to follow the arguments presented.

We have re-organized the paper.

Regarding the electrochemical ferricyanide reduction measurements, there is no clear explanation as to why the given applied potentials were chosen. S Fig 14 does not explain this as it is just a representation of the electrochemical setup. In addition, it would be beneficial to incorporate cyclic voltammograms of ferro/ferricyanide in biotic and abiotic samples in the same figure. Further discussion should follow regarding the R and O peaks in both scenarios. Other information such as CV scan rates would be helpful.

We now state in the paper our specific rationale for the oxidation and reduction potentials: For complete and quick bulk oxidation and reduction (< 20 min, Supplementary Fig. 10 b & c), we biased electrodes significantly more positively than the oxidation (+0.5V) or negatively than the reduction (-0.3V) peaks. Potentials closer to the peak potentials could be used, but conversion efficiency would suffer, and higher voltages can generate unwanted reactive species. Supplementary Figure 10 shows a cyclic voltammogram of Fcn(O/R) with the R and O peaks and we show a CV in both biotic and abiotic conditions (we saw no peak differences). Although the CV's are not informative, we show chronoamperometry measuring reduction of ferricyanide by cells and ferricyanide without cells, in Supplementary Figure 7d. Since the cells' reaction with ferricyanide and pyocyanin takes some time (100's to 1000's of seconds) and a cyclic voltammogram captures the current at each voltage for much shorter times (seconds or faster), the CA's are more informative of the interactions between cells and Fcn (O). We provide the CV scan rates in the Methods.

Appropriate use of statistics and treatment of uncertainties

The use of statistics was appropriate.

Conclusions: robustness, validity, reliability

The data presented in Figure 1 and SI Figures 11, 12 indicate that addition of a terminal electron acceptor is needed to reach the highest expression of SoxR-regulated genes in the presence of pyocyanin.

However, the conclusion that this mechanism driving this amplification of the pyocyanin response is indeed interaction of ferricyanide and pyocyanin with the respiratory ET chain as depicted in Figure 1b is not adequately supported. This mechanism is unsupported because of the following reasons:

The claim that pyocyanin can be re-oxidized by quinone components of the electron transport chain machinery (pg 4) is not supported by the reference indicated. Reference 31 does not discuss pyocyanin.

Thanks to the reviewer for this note. We do not conclude that the mechanism is through electron transport chain interactions. This is simply a hypothesis for further study, as ferricyanide is a known effector of electron transport (e.g., the native enzyme, ferricyanide dehydrogenase, is named as such even though the existence of ferricyanide in the native *E. coli* environments is unlikely). We concede in hindsight, that in the paper this may not be clear, and have revised the manuscript to clarify. Reference 31 (32 in new version) presents the hypothesis that several redox cycling drugs (MV/PQ, PMS, etc) work through re-oxidation by the quinone components of the electron transport chain. Although the reference does not specifically mention pyocyanin, it is a known redox cycling drug that activates SoxR; we inferred this and it is within the context of our mechanistic hypothesis. We reword this for clarity; we *used* this information to *drive our* experiments rather than make conclusions about the mechanism. Additionally, we modified our schematics and figures to not show direct interaction between Pyo and Fcn (O) in order to avoid confusion.

The claim that experiments under anaerobic conditions kept SoxR activation at low levels, below toxicity, (pg 4) is unsupported by SI Figure 12. SI Figure 12 does not address the level of SoxR activation or the overall cell viability under anaerobic conditions.

By low toxicity, we are referring to pyocyanin concentration, not SoxR activation. High levels of pyocyanin can be toxic to cells, but the amount we use does not seem to cause excessive cell death (SI Fig 6b).

If the mechanism is as indicated, then using 5 mM Fcn(R/O) should enable the effective pyocyanin (O) concentration to be 5mM instead of 5 μ M, i.e. a 1000-fold increase. Why there is only a ~17 fold increase in phiLOV expression?

Both pyocyanin and Fcn(R/O) likely interact with more than one component in the cell, thus one electron given to Fcn(O) would not necessarily be one electron that was taken from a pyo(O) molecule. But again, this is an hypothesis of the mechanism that would require extensive further study. Additionally, there are limits to induction, protein production, metabolism, etc that would come into play after a certain level of phiLOV is produced. This has been a theme of our lab for many years; heterologous protein expression is limited by the metabolic potential of the cells and is not linear with mRNA, promoter strength, etc.

The cell growth data presented in Supplementary Figure 8 is very unusual and suggests that growth is being altered under conditions that enable electronic control. Specifically, anaerobic growth of *E. coli* on glucose does not typically show the two phases presented in SFig 8a and the final OD is usually much higher, e.g. PLoS ONE 4(2): e4432. doi:10.1371/journal.pone.0004432.

We agree completely with the reviewer. There are likely many components of the cell physiology that are effected, but most importantly, we amplify gene expression from an individual promoter and show that this enables phenotypic change of the actuated cells AND that we can signal genetic changes in neighboring cells not otherwise affected by the electronics. The growth of the cells is altered; also the level of acetate secreted is altered. We have omitted the OD data in the revised version as the growth of the electrogenetically actuated cells is not particularly relevant to the desired outcomes. The effects on acetate, however, are more relevant and we've included them in the Supporting information (SFig. 6a). The reasons for this reduced acetate are not known, but we could speculate this is due to the added availability of electron transfer to Fcn and Pyo, which under "normal" anaerobic conditions can be limited, resulting in acetate overflow. Since this is speculative and peripheral, we have not addressed it in detail in the revised work.

Lastly, SFig 8a indicates that the the combination of redox mediators and pTT03 yields a growth defect. This change in growth matters because if cell growth is slowed, then the rate of protein dilution due to cell growth goes down and the steady-state protein concentration goes up. In short, the conditions of electronic control could slow growth and thus lead to increased fluorescence protein levels.

While the reviewer is certainly correct, the difference in growth rate and the impact this would have on the measurements of fluorescence is significantly lower than the fold increases observed. As noted above, we've removed the growth data to avoid confusion.

The claim that PMS generates comparable results is not supported by a figure or other data (pg 6).

We show PMS experiments in SI Figure 8 c & d. It was indeed our mistake that we did not reference this previously in the manuscript and have fixed this error.

Under anaerobic conditions, *E. coli* will readily ferment glucose, so electrons do not necessarily have to be entering the respiratory chain under these conditions. To directly implicate aspects of the electron transport chain in the effect seen here, the authors would need to show that quinone-deficient mutants, for example, do not show this same amplification. Also, showing that you get the same effect with a non-

fermentable electron donor would also significantly help implicate respiration. In short, while the data in the manuscript suggests that respiration is involved, the data is not strong enough to exclude alternative hypotheses and conclude that the mechanism is respiration-driven.

As the reviewer suggested, alternative electron acceptors support our claim of interaction with the electron transport chain. We actually did have this supporting data in the original manuscript, but it was buried. Our data (SI Fig 8b) shows nitrite/nitrate dependence. Further experiments would help with the mechanistic underpinnings, but as noted above, we have tried to embrace the vast quantity of published literature on oxidative stress and subsequent genetic responses elicited in *E. coli*.

Suggested improvements: experiments, data for possible revision
Please see section above.

References: appropriate credit to previous work?

As noted previously, the manuscript does not place the work appropriately in context because it fails to reference key areas of related work. The introduction needs to address the successes and shortcomings of using redox mediators by themselves to actuate cellular behavior, e.g. unbalanced fermentation, electrosynthesis; this is a significant literature, so reviews are helpful. Additionally, there are more sophisticated examples of using redox mediators and synthetic biology to electrochemically actuate behaviors, e.g. biosynthesis, gene expression, and this work needs to be cited.

As mentioned above, we have rewritten the manuscript to place our contributions in the correct context and address the successes and shortcomings of previous work as the reviewer suggests.

Clarity and context: lucidity of abstract/summary, appropriateness of abstract, introduction and conclusions

The manuscript lacks clarity because of repeated use of imprecise or incorrect words. For example, the authors refer to the 'gene amplification' multiple times (pg 7 line 6), when they are really referring to an increase in protein concentration, not the number of genes (DNA). On a more substantive issue, the authors describe using a degradation tag to switch gene expression "OFF" (pg 2, paragraph 1). This description is incorrect. A degradation tag does not significantly affect the process of gene expression (DNA->RNA->protein). Rather, it decreases the lifetime of that protein, so that the protein concentration more closely resembles the dynamics of gene expression and is less affected by the rate of cell growth. While the authors eventually address this point (pg 10), the initial discussion has already led the reader astray.

We have rewritten the manuscript to clarify the wording as correctly pointed out by R2.

Minor comments:

The title is far too broad. It should specify the organism and the use of redox mediators in combination with synthetic genetic circuits

We have changed the title to be more specific – “Electronic control of gene expression and cell behavior in *Escherichia coli* through redox signaling”.

The first two paragraphs of Results section seem more suitable for the Introduction section.

We have placed those two paragraphs in the Introduction section.

The statement that this work opens the doors for engineering cells that make decisions based on electronically relayed information (Discussion), i.e. electromagnetic, is incorrect. The field of optogenetics is all about engineering cells so that their behavior can be regulated by light. The authors cite this work in the introduction, so this statement is quite puzzling.

We recognize this error and have rewritten to be more precise.

Reviewers' Comments:

Reviewer #1 (Remarks to the Author)

The authors have satisfactorily addressed all of our concerns as well as those of the other reviewers.

Reviewer #2 (Remarks to the Author)

In this revised manuscript, Tschirhart et al. robustly characterize an electrogenetic device that uses minimal genetic re-wiring to enable electronic control of gene expression. The revised manuscript has addressed my three major concerns. It is more circumspect about describing the underlying mechanism and more focused on the characterization of the device. The language is much more precise, and the organization of the results and the SI is logical and streamlined. Overall, the revised manuscript offers a compelling and extensive characterization of a simple, new device for electronically regulating gene expression. The claims are well supported and will be of interest to a broad audience.